# Survival Outcomes and Predictors for Patients who Failed Chemoradiotherapy/Radiotherapy and Underwent Salvage Total Laryngectomy

**DOI:** 10.3390/ijerph18020371

**Published:** 2021-01-06

**Authors:** Ming-Hsien Tsai, Hui-Ching Chuang, Yu-Tsai Lin, Tai-Lin Huang, Fu-Min Fang, Hui Lu, Chih-Yen Chien

**Affiliations:** 1Department of Otolaryngology, Kaohsiung Chang Gung Memorial Hospital and Chang Gung University College of Medicine, Kaohsiung 833, Taiwan; b9302094@cgmh.org.tw (M.-H.T.); entjulia@cgmh.org.tw (H.-C.C.); xeye@cgmh.org.tw (Y.-T.L.); luhui88@cgmh.org.tw (H.L.); 2Kaohsiung Chang Gung Head and Neck Oncology Group, Cancer Center, Kaohsiung Chang Gung Memorial Hospital, Kaohsiung 833, Taiwan; victorhtl@yahoo.com.tw (T.-L.H.); fang2569@cgmh.org.tw (F.-M.F.); 3College of Pharmacy and Health Care, Tajen University, Pingtung County 907, Taiwan; 4Department of Hematology and Oncology, Kaohsiung Chang Gung Memorial Hospital and Chang Gung University College of Medicine, Kaohsiung 833, Taiwan; 5Department of Radiation Oncology, Kaohsiung Chang Gung Memorial Hospital and Chang Gung University College of Medicine, Kaohsiung 833, Taiwan; 6Institute for Translational Research in Biomedicine, Kaohsiung Chang Gung Memorial Hospital, Kaohsiung 833, Taiwan

**Keywords:** lymphovascular invasion, surgical margin, prognosis, laryngeal cancer, hypopharyngeal cancer

## Abstract

Background: To assess the presence of adverse pathological features at the time of salvage total laryngectomy (TL) associated with oncologic outcome. Methods: Ninety patients with persistent/locally recurrent disease and who subsequently underwent salvage TL after definitive treatment by radiation alone (RTO) or concurrent chemo-radiation (CCRT) from 2009 to 2018 were retrospectively enrolled. Kaplan–Meier methods were used to estimate overall survival (OS), disease-specific survival (DSS), and disease-free survival (DFS). Results: Lymphovascular invasion (LVI), perineural invasion, positive margin, and stage IV disease were associated with worse survival in the univariate analysis. In the multivariate analysis, the presence of LVI and positive margin were both independent negative predictors in OS (LVI: adjusted hazard ratio (aHR) = 2.537, 95% CI: 1.163–5.532, *p* = 0.019; positive margin: aHR = 5.68, 95% CI: 1.996–16.166, *p* = 0.001), DSS (LVI: aHR = 2.975, 95% CI: 1.228–7.206, *p* = 0.016); positive margin: aHR = 11.338, 95% CI: 2.438–52.733, *p* = 0.002), and DFS (LVI: aHR 2.705, 95% CI: 1.257–5.821, *p* = 0.011; positive margin (aHR = 6.632, 95% CI: 2.047–21.487, *p* = 0.002). Conclusions: The presence of LVI and positive margin were both associated with poor OS, DSS, and DFS among patients who underwent salvage TL after failure of RTO/CCRT. The role of adjuvant therapy for high-risk patients after salvage TL to improve the chance of survival requires more investigation in the future.

## 1. Introduction

Hypopharyngeal and laryngeal squamous cell carcinoma are not rare cancers in Taiwan, with about 1100 and 730 newly diagnosed cases in 2016, respectively. The age-standardized rate of hypopharyngeal cancer and laryngeal cancer are up to 6.13 per 100,000 persons and 3.96 per 100,000 persons, respectively, in males according to the report of the Taiwan Ministry of Health and Welfare. In general, in patients with locally advanced tumors of hypopharyngeal and laryngeal cancers, total laryngectomy (TL) plays a role in the treatment modality. In the last decade, organ preservation strategies have become popular treatments, since these laryngeal preservation protocols have reported similar overall survival rates to primary TL [1,2]. However, there are some patients who need surgical salvage for persistent or recurrent disease following radiotherapy alone (RTO) or concurrent chemoradiotherapy (CCRT). TL remains the recommended surgical procedure after radiation failure if the partial resection is not possible [3,4,5]. Currently, data are still limited regarding the survival impact of clinical and pathological findings after surgical salvage among these patients. The aim of this study is to determine the factors that influence the clinical outcomes among patients who have undergone salvage TL after primary treatment failure or recurrence at the primary site.

## 2. Materials and Methods

### 2.1. Study Population

Ninety consecutive patients with laryngeal/hypopharyngeal cancer who had undergone salvage TL with or without neck dissection were retrospectively enrolled in this study between October 2009 and May 2018. All of these patients were selected from the cancer database in Kaohsiung Chang Gung Memorial Hospital, Taiwan. Patients initially treated with primary RTO/CCRT and recurrent/residual tumors of their laryngeal/hypopharyngeal carcinoma were included. The immediate repair of the neopharyngeal defect was either by a primary closure or a free flap, such as an anterolateral thigh flap or anteromedial thigh flap. Patients were excluded if they had (1) salvage partial laryngectomy, (2) distant metastasis, (3) synchronous cancer, or (4) pathological complete response at the time of their salvage surgery. The treatment for these diseases was mostly based on the American National Comprehensive Cancer Network (NCCN) guidelines. The chemotherapy agent was cisplatin-based, and the radiation technique for all patients was intensity-modulated radiation therapy (IMRT). The primary radiation dose for all of our patients was between 70 and 74 Gy, with a conventional fractionated daily dose of 1.8 or 2 Gy. The status of each individual patient was evaluated and managed by a multidisciplinary team approach.

In total, 90 patients (86 men and four women) with a median age of 58.5 (range: 40–81) were included for analyses. Pathologists reviewed all of the surgical specimens with special emphasis on the presence/absence of perineural invasion (PNI), lymphovascular invasion (LVI), lymph node metastasis, extranodal extension (ENE), tumor grade, and the status of the surgical margins. Tumors were restaged according to the eighth edition of the American Joint Committee on Cancer Staging (AJCC).

### 2.2. Variables and Outcomes

Patients were retrospectively reviewed according to the following clinical characteristics: gender, age, primary cancer location, reasons for salvage surgery, cancer stage, and pathological features, including histological type, margin status, PNI, LVI, presence of nodal metastasis in neck dissection specimen, and presence of ENE, all of which were statistically analyzed for their influence on survival. The follow-up period in this study started from the date of radical surgery and ended at the date of death or last contact.

### 2.3. Statistical Analysis

Statistical analyses were performed using SPSS 20.0 software (SPSS/IBM, Inc., Chicago, IL, USA). The outcomes of interest included overall survival (OS), disease-specific survival (DSS), and disease-free survival (DFS). Univariate and multivariate Cox regression was performed to assess the association of the predictors of interest with survival outcomes. The hazard ratios (HRs) and their 95% confidence intervals (CIs) for each predictor were computed. The Kaplan–Meier method was utilized to estimate the probability of survival in each categorized factor, and the log-rank test was applied to examine the statistical significance of the factors. A two-tailed *p*-value < 0.05 was considered significant. This study was approved by the Medical Ethics and Human Clinical Trial Committees at Chang Gung Memorial Hospital (ethical application reference number: 202000562B0). Patients’ consent to review their medical records was not required by this hospital’s committees because the patient data remained anonymous in this study.

## 3. Results

A total of 90 patients were identified in this study, and their clinicopathological characteristics are summarized in Table 1. The median follow-up period was 24.1 months (range: 1.1–94.7). The most common tumor sub-site was the hypopharynx (*n* = 47, 52%), and the tumor location among the remaining patients was the larynx (*n* = 43, 48%). There were 48 patients (53%) who had undergone TL for a residual tumor after initial treatment, and 42 patients (47%) who had undergone salvage surgery for recurrent disease. The most frequent histologic type in this cohort, representing all but one patient, was squamous cell carcinoma; the other patient had spindle cell carcinoma. In tumor differentiation, most patients had moderately differentiated carcinoma (*n* = 83, 93%), followed by well-differentiated carcinoma (*n* = 3, 3%) and poorly differentiated carcinoma (*n* = 3, 3%). All hypopharyngeal and laryngeal cancers in this cohort were p16 negative tumors. Initially, there were 10 patients with clinical stage I disease, 10 patients with clinical stage II disease, 14 patients with clinical stage III disease, 27 patients with clinical stage IVA disease, and 29 patients with stage IVB disease before definite treatment. Seventy-eight patients received CCRT, and the remaining 12 patients received RTO as their initial treatment. The median period from the initial treatment to salvage TL was 5.9 months (range: 2.2–56.7). Fifty-nine patients had clinical stage IV disease (IVA, *n* = 35 and IVB, *n* = 24) in persistent or recurrent status before salvage TL, and 54 patients had pathological stage IV disease (IVA, *n* = 31 and IVB, *n* = 23) after salvage TL in our cohort. For pathologic T classification, this cohort included T1 (*n* = 7, 8%), T2 (*n* = 26, 29%), T3 (*n* = 14, 16%), T4a (*n* = 37, 41%), and T4b (*n* = 2, 2%), and four patients showed no tumor cells in the primary tumor area. The majority of patients (*n* = 74, 82%) received neck dissection in radical surgery. Nodal metastasis was present in 36 patients (49%), 25 of whom showed ENE (69%). N classification was noted as follows: N0 (*n* = 38, 42%), N1 (*n* = 8, 9%), N2a (*n* = 4, 4%), N2b (*n* = 1, 1%), N2c (*n* = 2, 2%), and N3b (*n* = 21, 23%). PNI and LVI were reported in 39% and 47% of patients, respectively. Five patients showed a positive surgical margin. Among these five patients, one patient had a positive margin over the carotid sheath, and died on postoperative day 47 due to carotid blowout. One patient had a positive margin over the mucosal of the trachea with submucosal spread, and re-irradiation, not CCRT, was advised due to poor cardiac function and renal function impairment of this patient. Three patients had a positive margin over the tongue base area, and adjuvant CCRT was advised for these patients. However, two of them suffered from persistent pharyngocutaneous fistula (PCF) postoperatively, with tumor relapse before adjuvant therapy started. However, these patients who had a positive surgical margin showed tumor recurrence within one year after salvage surgery, although two of them underwent additional adjuvant therapy followed by TL.

Among all patients, 75 (83%) underwent reconstruction with free flap transfer, including 74 anterolateral thigh flaps and one anteromedial thigh flap; 49 patients were reconstructed by the patch-on method, and another 26 patients were reconstructed by the tubing method. Major wound infection was defined as a wound condition, including PCF, free flap partial necrosis, and stoma wound necrosis, that should be debrided and managed in the operating room. The incidence of postoperative major wound infection was 34% (31/90).

Overall, two patients died during this hospitalization due to postoperative complications (2/90 = 2%). One patient had hypopharyngeal cancer (pT4bN0M0 with positive margin status), underwent salvage TL with free flap reconstruction for a persistent tumor after CCRT, and died on postoperative day 47 due to carotid blowout. The other patient, who had liver cirrhosis history (Child–Pugh classification A), underwent salvage TL for persistent supraglottic cancer (pT2N3bM0) with free flap reconstruction. This patient developed postoperative pneumonia and major wound infection and died on postoperative day 58 due to severe sepsis. In the follow-up period, tumor recurrence occurred in 38 (42%) patients after salvage TL. Relapsed areas, including local, regional, distant, both local and regional, and both regional and distant recurrence, occurred in eight, 12, 14, two, and two patients, respectively, in this cohort. At the time of the last follow-up, 52 (58%) patients remained disease-free, 32 (36%) patients had died of the disease, and nine (10%) patients had died of other diseases. All 90 patients in this cohort had a median OS of 43.8 months, a median DSS of 52.1 months, and a median DFS of 52.7 months.

Both univariate analysis and multiple analysis of potential pathologically adverse factors that showed an impact on overall survival are shown in Table 2. The presence of PNI (*p* = 0.033), presence of LVI (*p* = 0.001), positive surgical margin (*p* < 0.001), and stage IV disease (*p* = 0.046) were associated with significantly worse OS in univariate Cox regression analysis. In the multiple regression model, the presence of LVI (adjusted hazard ratio (aHR) = 2.537, 95% CI: 1.163–5.532, *p* = 0.019) and positive surgical margin (aHR = 5.68, 95% CI: 1.996–16.166, *p* = 0.001) were both significant adverse independent predictive factors of OS.

Regarding the DSS, we found that stage IV disease (*p* = 0.007), the presence of PNI (*p* = 0.006), presence of LVI (*p* < 0.001), positive surgical margin (*p* < 0.001), and female gender (*p* = 0.042) led to significantly worse DSS in univariate analysis. In the model of multiple regression analysis, the presence of LVI (aHR = 2.975, 95% CI: 1.228–7.206, *p* = 0.016) and positive surgical margin (aHR = 11.338, 95% CI: 2.438–52.733, *p* = 0.002) were both significant independent factors in DSS (Table 3). Patients with stage IV disease also had a worse prognosis in DSS, but this did not reach statistical significance in this model (aHR = 2.566, 95% CI: 0.999–6.594, *p* = 0.05; Table 3).

For DFS, we found that the presence of PNI (*p* = 0.004), presence of LVI (*p* = 0.001), and positive surgical margin (*p* < 0.001) were associated with significantly worse DFS in univariate analysis. In the model of multiple regression analysis, the presence of LVI (aHR = 2.705, 95% CI: 1.257–5.821, *p* = 0.011) and positive surgical margin (aHR = 6.632, 95% CI: 2.047–21.487, *p* = 0.002) were both significant independent factors in DFS (Table 4).

Patients with LVI at the time of salvage TL had worse oncological survival (Figure 1). Patients with LVI had shorter OS compared to those who did not have LVI (Figure 1A), with a median OS of 21.9 months vs. 72.2 months (*p* = 0.001). Patients with LVI had shorter DSS compared to those who did not (Figure 1B; *p* < 0.001), with a median DSS of 25.8 months for LVI-positive patients. Similarly, patients with LVI had shorter DFS compared to those who did not have LVI (Figure 1C; *p <* 0.001), with a median DFS of 12.6 months in the LVI group.

Patients who had a positive margin at the time of salvage resection had a statistically significantly (*p* < 0.001) shorter OS (7.2 months) compared to those with a negative margin (46.2 months; Figure 2A). Patients who had a positive margin upon salvage resection also had a significantly shorter DSS (with a median of 7.2 months) than patients with a negative margin (*p* < 0.001; Figure 2B). Similarly, patients with a positive margin upon salvage resection had poorer DFS (median 5.1 months) than those who had a negative margin (*p* < 0.001; Figure 2C).

The results demonstrate that the presence of LVI and positive surgical margin were the independent negative prognostic factors of OS, DSS, and DFS among patients who underwent salvage TL after failure of their primary radiation or chemoradiotherapy. The survival rates based on LVI and surgical margin are shown in Figure 1 and Figure 2.

## 4. Discussion

Multiple factors affect the prognosis in patients with laryngeal/hypopharyngeal cancer, most importantly the clinical TNM classification: in patients with increased T and N classifications and with the presence of distant metastases, the survival rate decreases. Boukovalas et al. [6] demonstrated that the presence of postoperative complications after primary/salvage TL in laryngeal/hypopharyngeal cancer was associated with poor oncologic outcomes. Several histopathologic features have been reported to be predictive of prognosis, with the presence of ENE being the most important sign of an unfavorable outcome [7,8]. Surgical margins status, patterns of invasion, and the presence of PNI and/or LVI may also impact survival in patients with laryngeal/hypopharyngeal cancer [9,10,11].

Even though organ preservation therapy is the most common treatment modality in laryngeal and hypopharyngeal cancers, 21–62% of patients still need salvage surgery for persistent or recurrent disease following radiotherapy or chemoradiation failure [12].

LVI is a common risk factor for locoregional recurrences and poor survival chances in patients with head and neck cancer [13]. Scharpf et al. [14] studied a cohort of 147 patients who received salvage partial laryngectomy or TL with persistent or recurrent laryngeal cancer and demonstrated that sarcomatoid pathology, presence of LVI, and an advanced initial stage are associated with inferior DFS. Basheeth et al. [15] concluded that, among 75 patients who had undergone primary and salvage laryngectomy, positive nodal disease and LVI significantly affected survival. In our current series of patients with salvage TL, 47% (42/90) of our patients had the presence of LVI. A significant inferior oncologic outcome, including OS, DSS, and DFS, was found in patients with LVI upon univariate and multivariate analysis. The persistence of LVI in tumor behavior even after RTO/CCRT revealed the possible radioresistance among these tumors.

Positive surgical margin is one of the most important factors in the prognosis among patients with HNSCC, according to previous studies. Jacques Bernier et al. [16] revealed that a microscopically involved surgical margin is a negative independent prognostic factor in HNSCC patients. Van der Putten et al. [17] studied a series of 120 patients who had undergone salvage TL and showed that only a positive resection margin was associated with worse DSS. Wulff et al. [18] conducted a retrospective analysis of 142 patients who had undergone salvage TL in tertiary centers in Denmark and showed that patients with tumor-free margins had better five-year OS and DSS. In our current series of patients with salvage TL, five patients had a positive margin after salvage TL; although one patient died from postoperative wound complications, the tumors relapsed in the other four patients during the first year of follow-up.

Clinically, these results suggest that patients who have adverse pathological features at the time of salvage surgery, including positive margin status and presence of LVI, need to be considered for more adjuvant therapy, although there is no clinical trial with level I evidence to support the benefit of this adjuvant therapy.

Interestingly, the variable of tumor location, which was expected to be a significant predictor, was not found to have a significant impact in this series. Hypopharyngeal cancer is known to have the worst prognosis among all head and neck cancers, despite the recent improvements in chemotherapy, radiation, and surgical technique [19,20]. This may come from the inclusion bias in the present series. Our patients with unresectable persistent/recurrent hypopharyngeal cancer after primary organ preservation therapy did not have the chance to undergo a salvage surgical procedure. Finally, there are limitations in this study. It is a retrospective study, and all patients underwent surgical procedures at a single institution performed by different head and neck surgeons. It is therefore vulnerable to selection bias.

## 5. Conclusions

Salvage TL is the way to improve clinical outcomes and disease control in hypopharyngeal/laryngeal patients with persistent/recurrent disease following failure of treatment by RTO/CCRT. Our study showed that the presence of LVI and a positive surgical margin are both independent negative predictive factors in OS, DSS, and DFS among patients who had undergone salvage TL for laryngeal cancer/hypopharyngeal cancer. The role of adjuvant therapy, such as immunotherapy, target therapy, chemotherapy, re-irradiation, or a combination, for high-risk patients after salvage TL to improve the chance of survival requires more investigation in the future.

## Figures and Tables

**Figure 1 ijerph-18-00371-f001:**
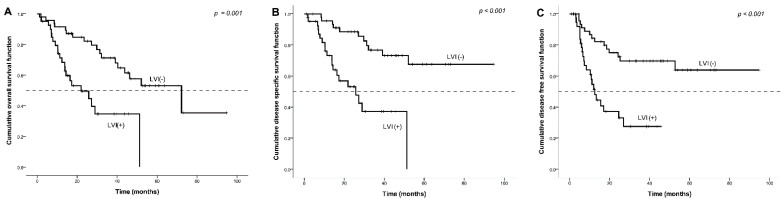
Kaplan–Meier survival curve by the presence of LVI. Impact of lymphovascular invasion (LVI) after salvage total laryngectomy TL on (**A**) overall survival, (**B**) disease-specific survival, and (**C**) disease-free survival.

**Figure 2 ijerph-18-00371-f002:**
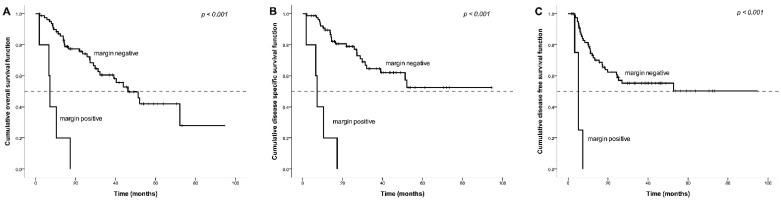
Kaplan–Meier survival curve by surgical margin groups. Impact of surgical margin after salvage total laryngectomy (TL) on (**A**) overall survival, (**B**) disease-specific survival, and (**C**) disease-free survival.

**Table 1 ijerph-18-00371-t001:** Clinicopathological characteristics of 90 patients who underwent salvage total laryngectomy.

Characteristics	Value (%)
Age, median (IQR) (range), year	58.5 (11) (40–81)
Male/female sex	86 (96)/4 (4)
Larynx/hypopharynx cancer location	43 (48)/47 (52)
Recurrent/Residual reason for salvage surgery	42 (47)/48 (53)
Histologic type	
WDSCC ^a^/MDSCC ^b^	3 (3)/83 (93)
PDSCC ^c^/Spindle cell carcinoma	3 (3)/1 (1)
Staging	
I/II/III	4 (4)/16 (18)/16 (18)
IVA/IVB	31 (34)/23 (26)
PNI ^d^	
Positive/Negative	35 (39)/55 (61)
LVI ^e^	
Positive/Negative	42 (47)/48 (53)
ENE ^f^	
Positive/negative	25 (69)/11 (31)
Margin	
Positive/Negative	5 (6)/81 (90)
not available (pT0)	4 (4)
Wound closure	
primary closure	15 (17)
free flap reconstruction	75 (83)
Design of free flap reconstruction	
patch on	49 (65)
tubing	26 (35)
Recurrence/no recurrence	38 (42)/52 (58)
Recurrent location	
Local	8 (21)
Regional	12 (32)
Distant	14 (37)
Both local & regional/Both regional & distant	2 (5)/2 (5)

^a^ Well-differentiated squamous cell carcinoma; ^b^ moderately differentiated squamous cell carcinoma; ^c^ poorly differentiated squamous cell carcinoma; ^d^ perineural invasion; ^e^ lymphovascular invasion; ^f^ extranodal extension, (*n* = 36).

**Table 2 ijerph-18-00371-t002:** Univariate and multivariate Cox regression analysis of prognostic factors associated with overall survival.

Factor	Univariable		Multivariable	
	HR ^g^ (95% CI)	*p*-Value	aHR ^h^ (95% CI)	*p*-Value
Age (≥58 vs. <58)	1.133 (0.607, 2.116)	0.695		
Sex (female vs. male)	2.846 (0.862, 9.398)	0.086		
Cancer location (hypopharynx vs. larynx)	1.543 (0.812, 2.932)	0.185		
Reason for salvage surgery (residual vs. recurrent)	1.134 (0.613, 2.098)	0.689		
Histologic type (WDSCC ^a^ or MDSCC ^b^ vs. PDSCC ^c^ or spindle cell carcinoma)	2.625 (0.359, 19.192)	0.342		
ENE ^d^ (positive vs. negative)	1.796 (0.636, 5.073)	0.269		
PNI ^e^ (positive vs. negative)	1.997 (1.056, 3.775)	0.033	1.354 (0.66, 2.776)	0.408
LVI ^f^ (positive vs. negative)	3.095 (1.59, 6.025)	0.001	2.537 (1.163, 5.532)	0.019
Pathological stage (Stage IV vs. Stage 0~III)	1.964 (1.013, 3.807)	0.046	1.715 (0.814, 3.613)	0.156
Margin (positive vs. negative)	10.112 (3.657, 27.959)	<0.001	5.68 (1.996, 16.166)	0.001
Wound closure (free flap reconstruction vs. primary closure)	1.956 (0.695, 5.504)	0.204		
Major wound infection (yes vs. no)	0.789 (0.371, 1.678)	0.538		
Design of free flap reconstruction (Patch on vs. tubing)	0.67 (0.349, 1.284)	0.227		

^a^ Well-differentiated squamous cell carcinoma; ^b^ moderately differentiated squamous cell carcinoma; ^c^ poorly differentiated squamous cell carcinoma; ^d^ extranodal extension; ^e^ perineural invasion; ^f^ lymphovascular invasion; ^g^ hazard ratio; ^h^ adjusted hazard ratio.

**Table 3 ijerph-18-00371-t003:** Univariable and multivariable Cox analysis of prognostic factors associated with disease-specific survival.

Factor	Univariable		Multivariable	
	HR ^g^ (95% CI)	*p*-Value	aHR ^h^ (95% CI)	*p*-Value
Age (≥58 vs. <58)	0.829 (0.411, 1.669)	0.599		
Sex (female vs. male)	1.872 (1.024, 3.425)	0.042	0.401 (0.065, 2.485)	0.326
Cancer location (hypopharynx vs. larynx)	1.382 (0.682, 2.802)	0.369		
Reason for salvage surgery (residual vs. recurrent)	1.097 (0.548, 2.198)	0.794		
Histologic type (WDSCC ^a^ or MDSCC ^b^ vs. PDSCC ^c^ or spindle cell carcinoma)	2.003 (0.276, 14.98)	0.486		
ENE ^d^ (positive vs. negative)	1.917 (0.599, 6.138)	0.273		
PNI ^e^ (positive vs. negative)	2.706 (1.333, 5.493)	0.006	1.685 (0.761, 3.73)	0.198
LVI ^f^ (positive vs. negative)	4.17 (1.944, 8.943)	<0.001	2.975 (1.228, 7.206)	0.016
Pathological stage (Stage IV vs. Stage 0~III)	3.163 (1.365, 7.329)	0.007	2.566 (0.999, 6.594)	0.05
Margin (positive vs. negative)	12.952 (4.572, 36.691)	<0.001	11.338 (2.438, 52.733)	0.002
Wound closure (free flap reconstruction vs. primary closure)	1.549 (0.542, 4.424)	0.414		
Major wound infection (yes vs. no)	0.707 (0.303, 1.65)	0.423		
Design of free flap reconstruction (patch on vs. tubing)	0.919 (0.424, 1.991)	0.83		

^a^ Well-differentiated squamous cell carcinoma; ^b^ moderately differentiated squamous cell carcinoma; ^c^ poorly differentiated squamous cell carcinoma; ^d^ extranodal extension; ^e^ perineural invasion; ^f^ lymphovascular invasion; ^g^ hazard ratio; ^h^ adjusted hazard ratio.

**Table 4 ijerph-18-00371-t004:** Univariable and multivariable Cox analysis of prognostic factors associated with disease-free survival.

Factor	Univariable		Multivariable	
	HR ^g^ (95% CI)	*p*-Value	aHR ^h^ (95% CI)	*p*-Value
Age (≥58 vs. <58)	0.927 (0.49, 1.754)	0.815		
Sex (female vs. male)	2.654 (0.81, 8.7)	0.107		
Cancer location (hypopharynx vs. larynx)	1.027 (0.747, 1.412)	0.87		
Reason for salvage surgery (residual vs. recurrent)	0.876 (0.464, 1.656)	0.684		
Histologic type (WDSCC ^a^ + MDSCC ^b^ vs. PDSCC ^c^ + spindle cell carcinoma)	1.879 (0.256, 13.776)	0.535		
ENE ^d^ (positive vs. negative)	1.71 (0.541, 5.41)	0.361		
PNI ^e^ (positive vs. negative)	2.604 (1.361, 4.983)	0.004	1.936 (0.943, 3.973)	0.072
LVI ^f^ (positive vs. negative)	3.278 (1.667, 6.447)	0.001	2.705 (1.257, 5.821)	0.011
Pathological stage (Stage IV vs. Stage 0~III)	1.952 (0.983, 3.875)	0.056		
Margin (positive vs. negative)	12.861 (4.033, 41.006)	<0.001	6.632 (2.047, 21.487)	0.002
Wound closure (free flap reconstruction vs. primary closure)	0.912 (0.42, 2.17)	0.912		
Major wound infection (yes vs. no)	0.863 (0.378, 1.973)	0.727		
Design of free flap reconstruction (patch on vs. tubing)	0.768 (0.373, 1.583)	0.475		

^a^ Well-differentiated squamous cell carcinoma; ^b^ moderately differentiated squamous cell carcinoma; ^c^ poorly differentiated squamous cell carcinoma; ^d^ extranodal extension; ^e^ perineural invasion; ^f^ lymphovascular invasion; ^g^ hazard ratio; ^h^ adjusted hazard ratio.

## Data Availability

The data presented in this study are available on request from the corresponding author.

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
