# Peer review of "Survival Outcomes and Predictors for Patients who Failed Chemoradiotherapy/Radiotherapy and Underwent Salvage Total Laryngectomy"

_ijerph, 2021, doi:10.3390/ijerph18020371_

Round 1
Reviewer 1 Report
The authors summarized the results of their retrospective analysis of patients who were treated with salvage total laryngectomy (and laryngopharyngectomy?) after recurrence or persistent laryngeal/pharyngeal cancer. All the patients had been treated with definitive radiotherapy or chemoradiotherapy. Multivariate analysis revealed that pathological findings such as LVI, positive surgical margins negatively affected outcomes. This finding is intriguing and seem valuable to prognosticate salvaged patients with recurrent/persistent disease after definitive RT/CRT. The authors summarized the manuscript well and the aim of the study was clear. However, there are several flaws.
1) Nearly half of the patients who experienced recurrence distant metastasis as an initial pattern of failure, i.e., 16 with distant metastasis and 2 with regional and distant metastasis out of the 40 patients in whom tumor recurrence occurred. This result implies substantial portion of the patients had systemic disease at the time of salvage surgery. Since it is difficult to infer the prognosis or risk of potential distant metastasis from pathological staging only, clinical staging which had been given before initial radiotherapy/chemoradiotherapy should be shown in the patient characteristics. And additionally, the duration between the initial radiotherapy/chemoradiotherapy and salvage TL should be presented because disease-free duration after initial treatment is also known to affect the ultimate treatment outcomes in patients with locally advanced head and neck squamous cell carcinoma.
2) The author should present their definition of surgical margin positivity, i.e., how many millimeters from the cut end were needed to deem the specimen as complete resected?)
3) The incident rate of major wound infection, which was 35%, seems very high. Please explain the detail of those events.
Reviewer 2 Report
This paper try to asses which clinicopathological factors affect the prognosis after salvage total laryngectomy or total pharyngo-laryngectomy. Authors analized data of a historicalcohort of 96 patients in the time frame2009-2018.
Results:
An overview of excluded patients for missing data should be given, anyway the sample size presented for the aim of the study is reasonable.
line 109:patients with compleate response:a histological assessment must always be donebefore such a mutilating operationt. If it was a funtional TL, please remove these patients from the study.
line 117-120:authors claim to have 5 positive margins and the managment post is not homogeneous, please explain what lead the choice to do CCRT, re-RT or nothing.
line 140-141:it is correctly presented the whole median survival, anyway it could be useful to show the KM estimated probability of survival (for the whole cohort) with 95%CI at specific target time points, such as 2- and 5- years, or build a table with this information.
From the data presented the rate of surgical complications with wound revision is known, a side analysis investigating variables associated with such complication could be interesting in this clinical scenario, any advantage of performing a flap rather than primary closure?
Round 2
Reviewer 1 Report
Dear authors,
The manuscript was corrected according to the suggestions for the reviewer.
Regards,